

# Common metrics of calibration for continuous Gaussian data and exceedance probabilities

Rita Glowienka-Hense[1], Andreas Hense[1], Thomas Spangehl[2], and Marc Schröder[2]

[1]Meteorological Institute, University of Bonn, Germany
[2]Deutscher Wetterdienst, Offenbach, Germany

**Correspondence:** Glowienka-Hense, Rita (rita.glowienka@uni-bonn.de)

**Abstract.** A framework of ensemble forecast verification tools is discussed which is founded on the concept of *information entropy*. It can be based on a common yardstick namely that of 'correlation'. With these measures calibration is deduced from the balance between ensemble sharpness and resolution. With the same units these features can be put into one diagram for continuous time series from Gaussian processes and exceedance probabilities, the latter usually tested with the reliability

term from the Brier score. The sharpness and resolution terms allow to use the same vocabulary of over- and underdispersion which is established for frequency histograms. The concept is based on the fact that mutual information (MI) of two Gaussian processes is directly related to Pearson's anomaly correlation. Further MI can be written as the Kullback-Leibler divergence of the conditional probability of observations given the model forecasts and the unconditioned observations. Thus the MI is a measure of resolution. The mean of the *UTILITY* defined by (Kleeman, 2002) is the corresponding measure of sharpness.

For Gaussian processes the mean *UTILITY* is very close to the ratio of ensemble mean variance to mean ensemble variance (*ANOVA*) which is the analysis of variance factor when time is taken as treatment. The ensemble spread score (*ESS*) (Palmer et al., 2006) is shown to be a measure of calibration if model and observed data are scaled with their respective means and standard deviations. For exceedance probabilities the resolution term of the divergence score (Weijs et al., 2010) is already defined as a MI term and it is here complemented with a mean *UTILITY* formed similarly to the resolution term but with

forecasts only. The entropy terms are then rescaled to the 'correlation' yardstick. The concept is applied to temperature data from the German project on decadal climate prediction, Mittelfristige Klimaprognose (MiKlip). It is shown that both over - and underdispersion can be found for the 2m temperature forecasts. Increasing ensemble sharpness of surface ocean temperature with lead year in the southern ocean hints at model-data inconsistencies at some locations in the ocean. Finally empirical orthogonal functions (EOF) of northern hemisphere annual mean surface temperature for ERA-40/ERA-Interim and MiKlip

retrospective hindcasts are determined. For both data sets the respective first EOF represents the low frequency temperature development. The time coefficients of the EOF are used to compare resolution and sharpness of continuous data and exceedance probabilities in one diagram.

*Copyright statement.* TEXT

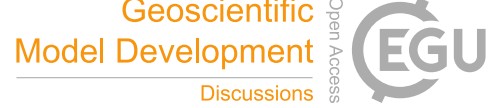



# 1   Introduction

Probabilistic forecast systems are intended to capture forecast uncertainty by the mean ensemble spread. In Murphy and
Winkler (1987) this is described through the analysis of the joint probability density or distribution of forecasts and obser-
vations. The calibration - refinement factorization of the joint density/distribution introduces calibration/reliability and re-
finement/resolution concepts. Whereas the likelihood-based rate factorization describes the sharpness/discrimination concept.
Gneiting et al. (2007) formulated that increasing sharpness leads to better forecasts as long as good calibration is kept. Good
calibration means a balance between sharpness and resolution. Potential predictability is associated with model sharpness.
Kleeman (2002) introduces the relative entropy also known as Kullback-Leibler (KL) divergence of a forecast ensemble and
model climate as a measure of sharpness of a single forecast he called this measure *UTILITY*. It is based solely on the model and
provides the additional information from the single forecast relative to model climate. We will introduce an average *UTILITY*
over many forecasts to determine a global measure of model sharpness. The mutual information (MI) (DelSole, 2004) between
observation and forecast is the corresponding information theoretical pendant for the resolution. It compares the observations
conditioned on forecasts with observational climate or expressed differently it is the relative entropy between the joint proba-
bility between observation and model and the corresponding marginal distributions. Ahrens and Walser (2008) use MI for the
interpretation of a contingency table instead of reliability diagrams which compare actual and predicted probabilities or ROC
(receiver operating characteristic) curves that show depending on a test statistic the true positive against false positive rates.
The resolution term of the divergence score (Weijs et al., 2010) has the same form and purpose. MI of two Gaussian processes
is a function of correlation only. Thus correlation is a measure of resolution for Gaussian processes. The mean *UTILITY* for
Gaussian processes (shown below) is on the other hand a function of the ratio of ensemble spread to total spread and the result
of an analysis of variance approach using the forecast time as treatment (*ANOVA*). Thus we have MI and *UTILITY* as measures
of calibration for Gaussian and categorical (e.g. temperature or rainfall exceedance probabilities) data. The transformation
formula of MI to correlation is used to scale MI to the $[0, 1]$ interval also for categorical data and a similar formula is used for
the *UTILITY*. This eases the interpretation of the MI and is done in statistical finance analysis (Dionísio et al., 2006). Palmer
et al. (2006) and Keller and Hense (2011) specified an ideal forecast ensemble such that the mean ensemble spread ($\sigma_e^2$) is
equal to the mean squared error of the model ensemble means and observations ($MSE$). The ensemble spread score (*ESS*)
compares mean model spread and forecast error. Here the *ESS* is used with normalized (subtracting the corresponding model
and observational means dividing by the standard deviations) variables as is done in (Kadow et al., 2016) because only then the
score is a measure of calibration as is shown here. For normalized variables the *ESS* is a function of correlation and *ANOVA*
and thus of resolution and sharpness. This bounds the *ESS* and we do not use it in the logarithmic form. The aim of this paper
is to determine calibration for Gaussian and categorical data using sharpness and resolution measures which are based infor-
mation theory. For continuous time series from Gaussian probability density function pdf these measures can be transformed
to correlation and *ANOVA* and similar transformations of the respective KL entropies are made for categorical data which are
of special interest for users. The use of sharpness and resolution instead of reliability and resolution for categorical data allows
the same interpretation as is done for continuous data. Thus the results for exceedance data and continuous time series can be





directly compared. In the next 3 sections the *ESS* in terms of correlation and *ANOVA*, the relation between mean *UTILITY* and *ANOVA*, as well as the formulations of resolution and sharpness for categorical data are derived. Applications show the balance between correlation and *ANOVA* and the associated *ESS* for anomalies of near surface air temperature (TAS) from 45 medium range climate reforecasts (1962-2012) obtained from the German medium range climate prediction project MiKlip (Marotzke

et al., 2016) at lead year 2, further how the development of sharpness at different lead times reveals initialization problems of the forecasts which obviously stem from the ocean initialization data and two EOF analysis of northern hemisphere TAS from ERA40/ERAinterim data and from 45 medium range climate reforecasts at lead year 2-5 (1962-2012). The first EOF of both analyses represent the Northern Hemisphere low frequency variation of temperature and the associated patterns in the observational and model data. From the EOF coefficients correlation, *ANOVA* and *UTILITY* are calculated for the continuous

time series together with the corresponding transformed resolution and sharpness for exceedance probabilities of the median and upper quartile.

## 2 Ensemble spread score dependence on Pearson's anomaly correlation and sharpness in terms of *ANOVA*

The ensemble spread score (*ESS*) compares mean model spread $\sigma_e^2$ and mean squared error between the ensemble mean and the observations (MSE).

$$ESS = \frac{\sigma_e^2}{MSE} \tag{1}$$

An *ESS* of 1 indicates that on average the squared distance between observations and ensemble means is the same as that between the ensemble members and the ensemble means. Deviations from this optimum value can result from differences in the standard deviations between model and observations or from a missing balance between model sharpness and resolution. The former differences can be directly found by comparing the standard deviations themselves. The actual issue of the *ESS* is

to be a measure of calibration. (Kadow et al., 2016) used the log-transform of the *ESS* ratio for MiKlip retrospective hindcast verification. Especially they standardized the model data with the square root of the total variance and the observations accordingly with its standard deviation. In this form the *ESS* compares sharpness and resolution. It is shown here that for standardized model and verification data the ESS can be written as a function of correlation and the ratio of ensemble mean to total variance. The latter is an analysis of variance (von Storch and Zwiers, 2001) with time as treatment which is called here *ANOVA* for

brevity. For any forecast variable $Y_{ij}$ with i counting the runs and j denoting time, the mean over runs and time $Y_{00}$ and the ensemble means $Y_{0j}$ are given by:

$$
\begin{aligned}
Y_{00} &= \frac{1}{J*nrun} \sum_{j=1}^{J} \sum_{i=1}^{nrun} Y_{ij} \\
Y_{0j} &= \frac{1}{nrun} \sum_{i=1}^{nrun} Y_{ij}
\end{aligned}
\tag{2}
$$





The total model variance $\sigma_t^2$, the mean ensemble spread $\sigma_e^2$ and the ensemble mean spread $\sigma_a^2$ are defined as follows:

$$\sigma_t^2 \equiv \frac{1}{J*nrun} \sum_j \sum_i (Y_{ij} - Y_{00})^2$$

$$\sigma_e^2 \equiv \frac{1}{J*nrun} \sum_j \sum_i (Y_{ij} - Y_{0j})^2$$

$$\sigma_a^2 \equiv \frac{1}{J} \sum_j (Y_{0j} - Y_{00})^2$$

$$\sigma_t^2 = \sigma_a^2 + \sigma_e^2$$

(3)

The total variance is the sum of the variance of the ensemble mean and the mean ensemble spread and the *ANOVA* is thus:

$$ANOVA \equiv \frac{\sigma_a^2}{\sigma_t^2}$$

(4)

5    If the observations $X_j$ and forecasts $Y_{ij}$ are standardized as follows:

$$\hat{X}_j = \frac{X_j - X_0}{\sigma_x}$$

$$\hat{Y}_{ij} = \frac{Y_{ij} - Y_{00}}{\sigma_t}$$

(5)

the variance of $\hat{X}$ and the total variance of $\hat{Y}$

$$\sigma_x^2(\hat{X}) = 1$$

$$\sigma_t^2(\hat{Y}) = 1$$

(6)

and thus the mean ensemble spread and the mean square error for the scaled variables become

$$\sigma_e^2(\hat{Y}) = 1 - ANOVA$$

$$MSE(\hat{X}, \hat{Y}) = \sum_j (\hat{Y}_{0j} - \hat{X}_j)^2$$

$$= ANOVA + 1 - 2*CORR*\sqrt{ANOVA}$$

$$ESS(\hat{X}, \hat{Y}) = \frac{1 - ANOVA}{ANOVA + 1 - 2*CORR*\sqrt{ANOVA}}$$

(7)

whereby CORR denotes the correlation between observations (or reanalysis) and ensemble means. The *ESS* thus forms a bridge between the deterministic ensemble mean correlation and the ensemble spread. In the ideal case the ensemble spread score is equal to one. Viewed with these scaled variables the following relations hold between model sharpness *ANOVA*, correlation and the shape of the rank histogram (RH). The latter is a standard method to test the reliability of an ensemble

15   forecast. In case of a flat RH, the ensembles are reliable and the uncertainty of the forecast is given by the ensemble spread.





U-shaped/inverse U-shaped RH indicate that the distance between ensemble mean and observations is larger/smaller than that between ensemble members and ensemble mean. This is called underdispersion/overdispersion because the ensemble spread underestimates/overestimates the forecast uncertainty.

$$
\begin{aligned}
ESS = 1 \quad & \left\{ \begin{array}{ll} CORR = & \sqrt{ANOVA} \\ ANOVA = & 0 \end{array} \right\} \quad \text{flat RH} \\
ESS > 1 \quad & CORR > \sqrt{ANOVA} \qquad \text{inverse U-shaped RH} \\
ESS < 1 \quad & CORR < \sqrt{ANOVA} \qquad \text{U-shaped RH}
\end{aligned} \tag{8}
$$

If the model has no sharpness, i.e. if $ANOVA = 0$ it is perfectly reliable but the forecast is indistinguishable from model climate. Thus there is always an ideal relation between model ensemble spread and scaled mean square error as the model finally loses its sharpness for long range forecasts. In case of ensemble sharpness and $ESS = 1$ there is a balance between ensemble sharpness and forecast error. The latter relation is tested in (Eade et al., 2014) who defined the index "ratio of predictable components"(RPC):

$$
RPC \geq \frac{CORR}{\sqrt{ANOVA}} \tag{9}
$$

The analysis of ensemble forecasts thus requires both correlation and sharpness which is equivalent to one minus the ratio of ensemble spread to total variance. The *ESS* dependence on correlation and *ANOVA* is displayed in Fig. 1. The grey band containing the optimal $ESS = 1$ is very large for low sharpness and gets rapidly smaller for increasing sharpness and correlation. A cross section at CORR ≥ 0.6 has two intersections with the gray band as the *ESS* relation has two solutions for ANOVA

for fixed CORR and *ESS* (7). The lower *ANOVA* values correspond to a broader band of acceptable balance between model and observation, whereas increasing ensemble sharpness demands a much closer correspondence between sharpness and correlation to attain the same statistical balance. Thus the information of good calibration between ensemble spread and forecast success can only be rated objectively with knowledge of the sharpness. In a statistical sense both intersections of the gray zone are equally well calibrated but the benefit is of course higher for large sharpness. The *ESS* and the *RPC* both classify the opti-

mal relation of correlation and *ANOVA* as $CORR = \sqrt{ANOVA}$ and both factors are one in this case. If $ESS < 1$ the model ensemble is underdispersive and has larger potential - than actual predictability. This can either be an initialization problem of too low spread and thus real underdispersion or the mismatch between model sharpness and correlation is due to the fact that the physics of the model are not fully correct. On the other hand if the $ESS > 1$ the model ensemble is overdispersive and the ensemble spread is larger than the distance between observations and ensemble mean. For this the most natural explanation

seems to be that either the observational data are too smooth or that there is too much internal model noise. The regions of underdispersion are however by far larger in our analysis.





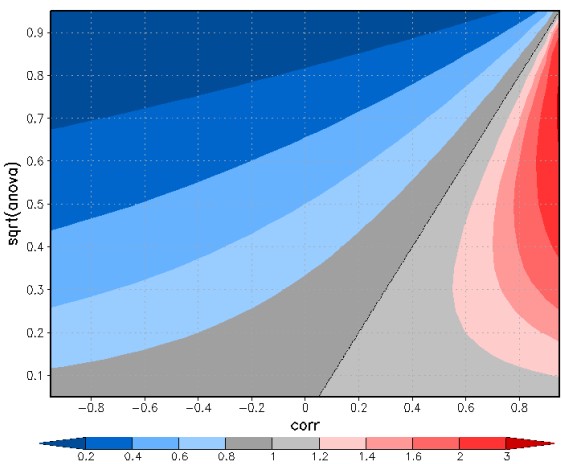

**Figure 1.** $ESS$ as function of square root of the anova and correlation.

## 3 *UTILITY* and MI and their relation to *ANOVA* and correlation

A further brick that fits into the concept is the *UTILITY* which is a measure of information of a single time step ensemble forecast proposed by Kleeman (2002). The single time step in this context can be a daily, a decadal or any other mean for a fixed date. The *UTILITY* is defined as the relative entropy between forecast ensemble $p(x)$ and the corresponding model

5   climate probability density function (pdf) $q(x)$.

$$UT = \int\limits_{-\infty}^{+\infty} p(x)log(p(x)/q(x))dx \qquad (10)$$

In the univariate case like a single model grid point, an area average or a mode amplitude and under the assumption of Gaussian distribution of the variable $Y$ it has the following form Kleeman (2002):

$$UT = \frac{1}{2}\left(ln\left(\frac{\sigma_t^2}{\sigma_{ej}^2}\right) + \frac{\sigma_{ej}^2}{\sigma_t^2} - 1\right) + \frac{1}{2}\left(\frac{(Y_{0j} - Y_{00})^2}{\sigma_t^2}\right) \qquad (11)$$

10   Kleeman (2002) decomposed the expression into the *signal* term containing the mean and the *dispersion* terms. The signal term is equal to the Mahalanobis distance and can be tested with the Hotelling-$T^2$ statistic (von Storch and Zwiers, 2001). Where the signal term gets small only the spread terms provide sharpness DelSole (2004). At this point ensemble forecasts are essential. *UTILITY* depends only on the forecasts as does the *ANOVA* for the mean predictability. It is thus a measure





of sharpness or potential predictability for a single forecast. *UTILITY* is a necessary but not sufficient condition for a good forecast. Averaging the *UTILITY* over many forecasts of a fixed forecast horizon leads to:

$$
\begin{aligned}
\frac{1}{J}\sum_j UT_j &= \frac{1}{2J}\sum_j ln\left(\frac{\sigma_t^2}{\sigma_{ej}^2}\right) + \frac{\sigma_e^2}{\sigma_t^2} + \frac{\sigma_a^2}{\sigma_t^2} - 1 \\
\frac{1}{J}\sum_j UT_j &= \frac{1}{2J}\sum_j ln\left(\frac{\sigma_t^2}{\sigma_{ej}^2}\right) + 1 - ANOVA + ANOVA - 1 \\
\frac{1}{J}\sum_j UT_j &\sim -\frac{1}{2}ln(1 - ANOVA)
\end{aligned}
\tag{12}
$$

The last relation is an approximation which holds exactly if the ensemble spread $\sigma_{ej}$ is constant for each forecast. In any case the mean *UTILITY* only depends on the ensemble and model climate spread and no longer on the mean. The relation between two processes can be measured with the mutual information. It is defined as relative entropy of the joint and the product of the marginal distributions of two processes:

$$
MI = \sum_{x \in A}\sum_{y \in B} p(x,y) log \frac{p(x,y)}{p(x)p(y)}
\tag{13}
$$

The average relative entropy and MI of two Gaussian processes have a similar form whereby the *ANOVA* is replaced by the squared Pearson anomaly correlation coefficient in case of MI.

$$
MI = -\frac{1}{2}ln(1 - CORR^2)
\tag{14}
$$

This underlines the joint role of *ANOVA* and correlation for ensemble forecast verification, while the former measures common information of the forecast ensemble, the latter quantifies the common information between the ensemble and the observation data set (in the following examples reanalysis data).

## 4 Mutual Information and mean *UTILITY* for categorical data

Now we want to extend the analysis of sharpness and resolution to categorical data and present a method that allows to keep the terminology from the continuous Gaussian case. Reliability diagrams are used to verify categorical data like limit exceedance. In a reliability diagram the probability of forecasts for a certain predefined event is compared to the observed frequency of that event whereby forcasted probabilities are grouped into bins. Weijs et al. (2010) presented the divergence score to quantify the mean resolution and reliability of multicategorical forecasts. Especially the resolution of the divergence score is the mean MI between model and observed categorical probabilities. A similar score based on MI has been presented by Ahrens and Walser (2008). MI can be transformed to a global correlation coefficient (GCC) using the relation between correlation and MI for two Gaussian processes (see equation (14)). This transformation is known from financial market analysis. For the resolution term





the observations are binned according to the binned forecasts as in the reliability diagram. Then the mean KL divergence is computed between the mean probabilities of these bins and the global mean observation probability. For the resolution term we adhere directly the proposition of Weijs et al. (2010) and Ahrens and Walser (2008).

$$
\begin{aligned}
RES_{DS} &= \frac{1}{N} \sum_{k=1}^{K} n_k D_{KL}(\hat{X}_k // \hat{X}) \\
&= \underset{k}{E} D_{KL}[(\hat{X}/Y_k) // \hat{X}] \\
&= MI(Y; X)
\end{aligned}
\tag{15}
$$

$E$ denotes the expectation value. The resulting estimate of the MI is then scaled to a global correlation coefficient (GCC). Thus we do not scale the score as proposed by Weijs et al. (2010) and Ahrens and Walser (2008). The GCC has the advantage to be limited to the range $[0, 1]$:

$$
GCC \equiv \sqrt{1 - exp(-2MI)}
\tag{16}
$$

Deviating from Weijs et al. (2010) the reliability of the categorical forecasts here is not determined from the KL divergence of oberved and model probabilities of the respective bins. Instead the mean *UTILITY* of the forecasts is used as a measure of sharpness as in the continuous case where the mean *UTILITY* can be scaled to ANOVA. Thus we can directly test the balance between sharpness and resolution as claimed in (Gneiting et al., 2007). To this end we use the same formalism as for the resolution term but for the model data.

$$
\widehat{UT} = \frac{1}{N} \sum_{k=1}^{K} n_k D_{KL}(\hat{Y}_k // \hat{Y})
\tag{17}
$$

The mean *UTILITY* which corresponds to (12) in the continuous case is then transformed to a $GAC$ (Global ANOVA Coefficient) with a similar transformation as used for the GCC (Eq (16)):

$$
GAC \equiv \sqrt{1 - exp(-2 * \widehat{UT})}
\tag{18}
$$

Now we have the same terms namely MI and mean *UTILITY* or resolution and sharpness as in the continuous case. And using the transformations to GCC and GAC (16),(18) the evaluation of reliability can be done in the same way as in the continuous Gaussian case. The GAC corresponds to the square root of the ANOVA in the continuous case. The reliability is optimal if

$$
\frac{GCC^2}{GAC^2} \left\{
\begin{array}{lll}
= & 1 & \text{optimal} \\
> & 1 & \text{overdispersive} \\
< & 1 & \text{underdispersive}
\end{array}
\right\}
\tag{19}
$$



Both MI and mean *UTILITY* (sharpness) are scaled quantities and we compare the information difference between model and observation and the difference between forecasts and model climate. As in case of the *ESS* the absolute values of GCC and GAC should be kept in mind because if both were too small the interpretation gets meaningless. This interpretation of existing methods is affected by the same issues associated with the data binning as do the Brier and the Divergence Scores. The interpretation of the actual numbers is however much more intuitive than the aforementioned scores and is comparable to the continuous case.

## 5 Data

The utilized model data are decadal hindcasts from the MiKlip decadal climate prediction system (Pohlmann et al., 2013), (Kadow et al., 2016). 45 runs are used, i.e. Baseline 1 with 10 low and 5 mixed resolution runs with nudging to ORAS4 (Balmaseda et al., 2013) ocean reanalyses anomalies and nudging to ERA-40/ERA-Interim in the atmosphere and 15 prototype runs each PG2LR and PS4LR with GECCO2 (Köhl and Stammer, 2008) and ORAS4 ocean full-field initialization respectively. Additionally comparisons are done with a 12 member ensemble of uninitialized historical runs (Giorgetta et al., 2013), which are however forced by greenhouse gases and volcanic forcing. The model is the Max-Planck-Institute Earth System Model (MPI-ESM-LR for low resolution and MPI-ESM-MR in case of the mixed resolution version). Details of the ocean component with the resolution of $1.5°/L40$ (LR version) can be found in Jungclaus et al. (2013). The atmospheric component is ECHAM6 (Stevens et al., 2013) and has a resolution of T63L47 in the LR version. The mixed resolution version is T63/L95 in the atmosphere and $0.4°/L40$ in the ocean and is able to simulate the quasi biennial oscillation (Scaife et al., 2014). The model data are compared to ERA-40 (Uppala et al., 2005) in 1961 to 1978 and to ERA-Interim (Dee et al., 2011) in 1979 to 2012.

## 6 *ANOVA* analysis of near surface temperature anomalies

In the following two examples are given where the information of ensemble sharpness aids the interpretation of the forecast issue. The balance of correlation and *ANOVA* is discussed which varies between over- and underdispersion. Underdispersion indicates regions where the model has potential predictability but does not or only partly fit the observations. The interpretation of overdispersion is less clear (see below). The sensitivity of 2m temperature (TAS) medium range climate predictions to the deep ocean initialization is demonstrated with *ANOVA* analysis for different lead times.

### 6.1 Relation between correlation and *ANOVA*

Correlation with ERA-40/ERA-Interim, *ANOVA* and *ESS* for TAS lead year 2 of MiKlip retrospective hindcasts (Fig. 2) and historical runs (Fig. 3) are compared. These second moment statistics are especially sensible to ensemble size. Thus here the ensemble hindcasts of four MiKlip experiments are sampled. Together this are 45 runs and 54 hindcast years (1962-2012, lead year 2). (Eade et al., 2014) discuss the overdispersion of TAS especially over the North Atlantic, Northern Africa and Eastern Europe. Sheen et al. (2017) find the same phenomenon of overdispersion. The latter use high resolution (about 60km



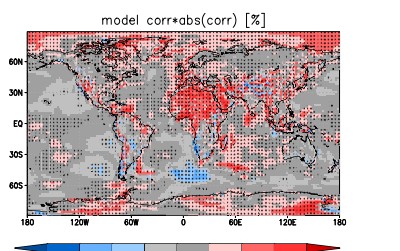
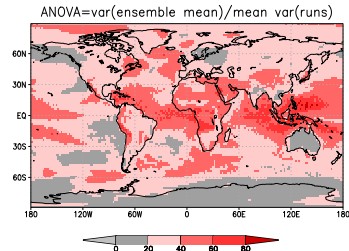
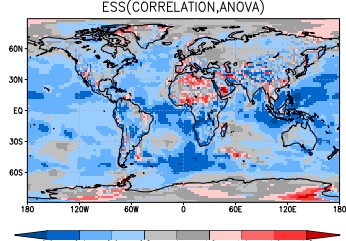

**Figure 2.** TAS correlation, *ANOVA* and *ESS* for lead year 2 for combined 45 runs of MiKlip phase I and ERA-40/ERA-Interim. Correlations above the $5\%$ significance level according to local Student-T statistic are stippled.

in the atmosphere) reforecasts. The MiKlip ensemble system also tends to overdispersion in regions of high correlation. The correlations are shown as $CORR * \|CORR\| * 100$ giving explained variances but keeping the sign to be comparable to the *ANOVA* results. Their relation is given by the *ESS*. If we first look at Africa we see large regions of high correlation and *ANOVA* mainly north of the equator. The largest correlations are however in the regions of relatively low *ANOVA* and thus

the *ESS* is well above 1.4 in the regions surrounding the Sahel Zone. South of the equator on the African continent, over the eastern Atlantic and Indian Ocean the model ensemble is mainly underdispersive. Over the oceans the model ensemble reaches *ANOVA* values of above 30% over relatively large regions but this does not lead to good forecasts. The main difference between hindcasts and historical runs is that the degree of overdispersion ($ESS > 1$) in regions of good correlation is much reduced for the historical runs. We suspect the model drift during the first years of the hindcast runs leads to the additional model noise

as they are sampled from 54 (1962-2012) different initializations which might lead to different model drifts. DelSole (2004) discuss that a backround random noise field can hide a highly predictable component. Regions of low *ESS* with *ANOVA* larger than the corresponding correlations squares are only found if the latter are below about 60%. This supports the hypothesis that in these regions the underdispersion could be a problem of the model or low frequency initialization data like ocean temperature and salinity (see next section).

**7   Ocean surface temperature affected by data errors from the ocean**

Generally the *ANOVA* of forecast variables decreases with increasing lead time, but this must not be true for single variables like TAS. Positive *ANOVA* is a necessity for a successful forecast but if it is not associated with positive correlation between model and observational data it is an indication of possible model or data problems. The ability for successful medium range climate forecasts relies on the long memory of the ocean which can provide a low frequency forcing especially for TAS over

the oceans. In the MiKlip retrospective hindcast system two ocean reanalysis data sets have been used Prototype GECCO2 and Prototype ORAS4. Over the ocean both hindcast sets show regions with systematically increasing ensemble sharpness with lead time. These regions however differ between the data sets. Figs. 4 and 5 show differences of *ANOVA* for lead year 2





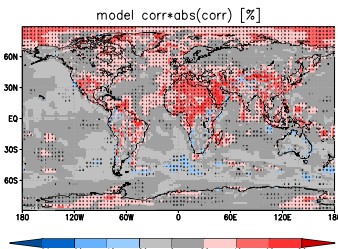
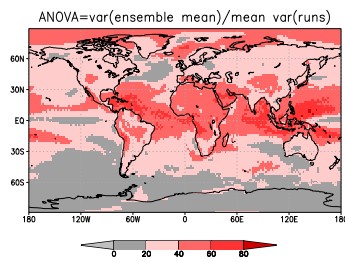
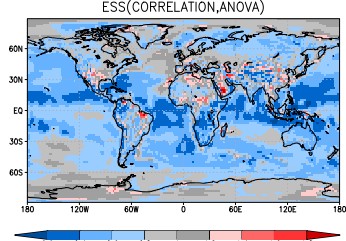

**Figure 3.** Same as Fig. (2) but for historical runs.

and higher lead years. The contours show the sharpness (*ANOVA*) at lead year 2 and the differences are shaded. Red areas are those where the spread of the ensembles is lowest in lead year 2 as expected. The *ANOVA* is calculated over the hindcast years 1970-2012, which is the common period for all lead horizons. In the PG2LR runs there is a blue shaded region, where *ANOVA* increases with lead year over the southern ocean (Fig. 4) in the difference field between lead year 2 and 3 to the west of southern

South Africa. Till lead year 6 this region expands to the southern ocean and over half of the hemisphere (not shown) and stays coherent till lead year 10. Thus there is a systematic difference of the forecasts for the same years 1970-2012 depending on the time that has passed since the initialization. The hincasts based on PS4LR show sharpness increase at a somewhat different location namely west of the Antarctic Peninsula here the maximum is reached in lead year 6, then the extra coherence decreases to zero at lead year 10 (not shown). The PG2LR runs show a similar phenomenon in the eastern North Atlantic (Fig. 5) where a

small region of coherence increase develops near Great Britain and Spain and then extends and moves southward to the western African coast where it expands eastward to the central Atlantic till lead year 10.

## 8   Northern Hemisphere TAS EOF comparison of continuous correlation and *ANOVA* with exceedance estimates of GCC and GAC

The Northern Hemisphere EOF 1 of TAS averaged over lead years 2-5 from 45 MiKlip hindcast runs is compared to the corre-

sponding EOF 1 based on ERA-40/ERA-Interim reanalysis 4-year running averages to emphazise the long term development. The EOF analyses are based on the correlation matrix. The model EOF have been derived from the average correlation matrix of all runs. Comparable results are attained if the EOF are derived from the ensemble mean time series and the coefficients are projected onto the modes of the ensemble mean. The EOF 1 pattern loadings from annual means (Fig. 6) are positive over the hemisphere for model data while the ERA-40/ERA-Interim EOF 1 loadings are negative over the Pacific and positive over

most of the rest of the hemisphere imaging the recent cooling over the northern mid latitudinal Pacific (Trenberth and Fasullo, 2013). A similar analysis of the global surface temperature has been performed by (Zorita et al., 2005) to find the local pattern of global temperature change. Here the analysis is restricted to the northern hemisphere because the models with different



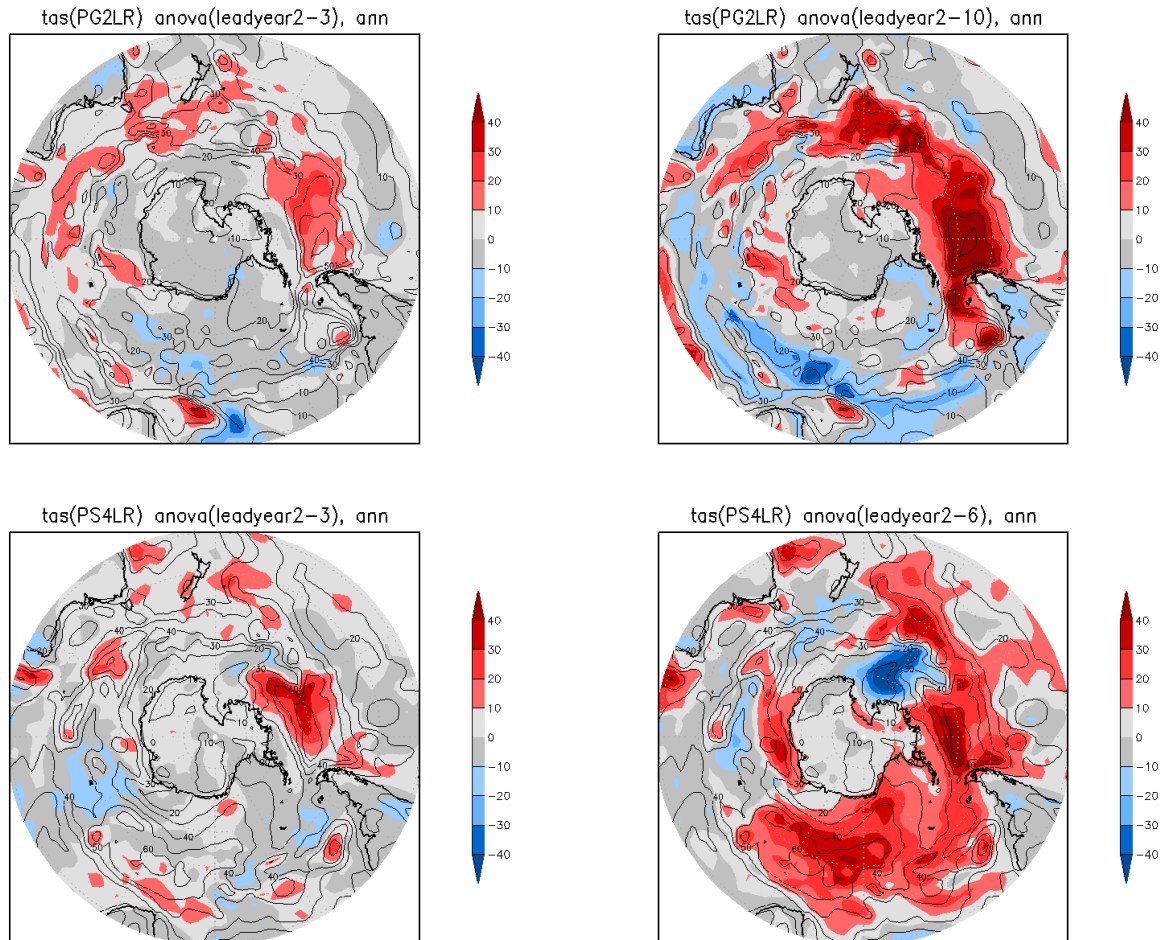

**Figure 4.** Annual mean TAS *ANOVA* differences for the southern hemisphere for PG2LR runs (upper row) lead years 2-3 (left) and 2-10 (right) and for PS4LR runs (lower row) lead years 2-3 (left) and 2-6 (right). Blue areas are those where model sharpness contrary to intuition increases with lead time due to information from the deep ocean. The areas differ between the Prototype runs.



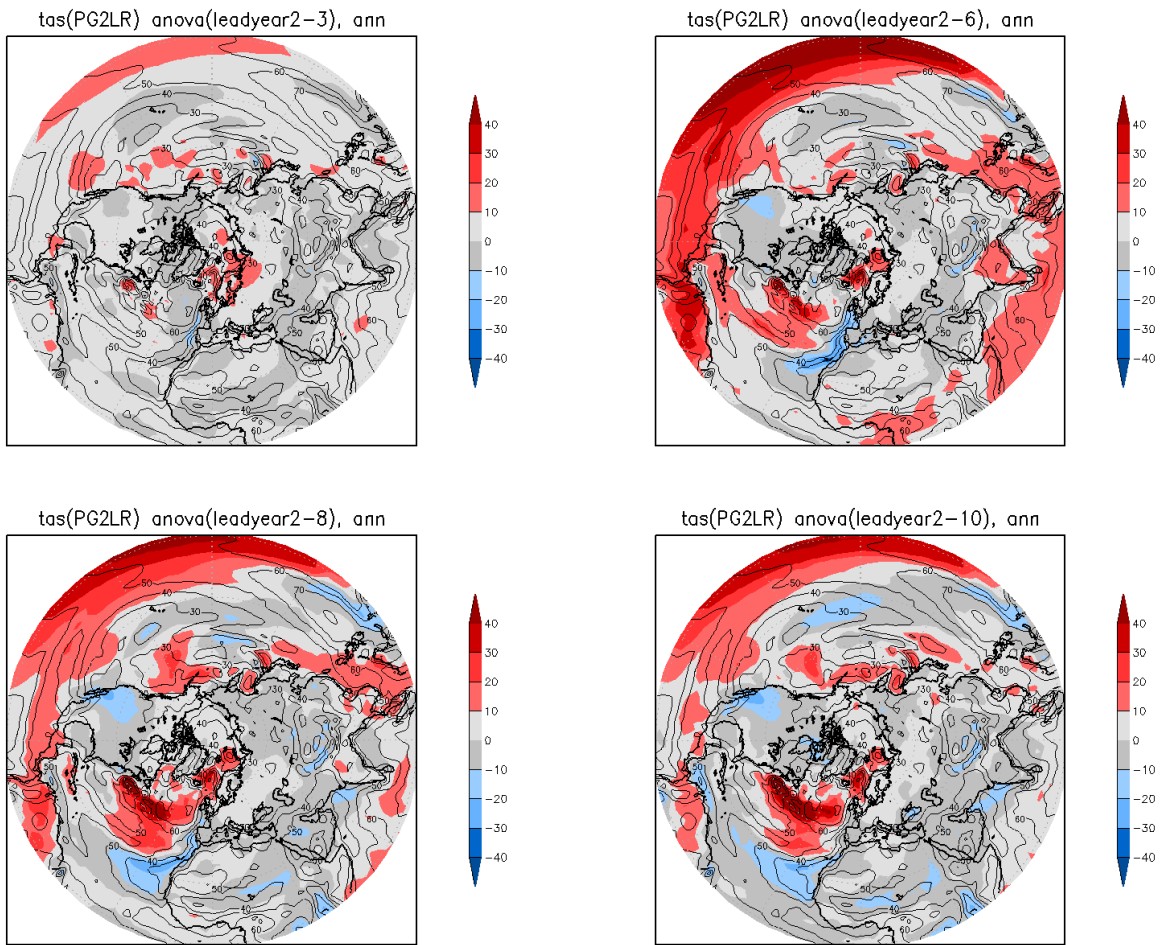

**Figure 5.** Annual mean TAS *ANOVA* differences for the northern hemisphere for PG2LR runs, upper row: lead years 2-3 (left), 2-6 (right),lower row: lead years 2-8 (left) and 2-10 (right). Blue areas are those where model sharpness contrary to intuition increases with lead time possibly due to an emerging signal from the deep ocean. Along the western European coast the coherence increases with lead time and moves to the African coast for lead years 8 to 10.





ocean temperature initialization differ strongly over the southern ocean. The model EOF loadings over the Pacific are smaller than in other regions but still positive. The typical colder ocean warmer land pattern appears if the modes are scaled with the local standard deviations (not shown). The EOF expansion coefficient time series have very similar low frequency behavior. Fig. 6 shows the ERA-40/ERA-Interim and ensemble mean time series (scaled with their maximum times 100) together with the

*UTILITY* in GAC units. The *ANOVA* and the mean *UTILITY* form the indistinguishable horizontal lines at 80% in the figure. As discussed the *UTILITY* is smaller where the EOF expansion coefficients are close to zero, but due to the small ensemble spread relative to climate variance it stays above 40%. Comparable analyses have been done for multi year means of single months resulting in similar EOF 1 patterns and expansion coefficients (not shown). Single year analyses lead to ERA-40/ERA-Interim and model modes where even EOF 1 and EOF 2 are not well separated in winter presumably due to the large Pacific North

American (PNA) - and North Atlantic Oscillation (NAO) signals (not shown).

The resolution and sharpness (Fig. 7) for the median and upper quartile exceedance of the EOF1 coefficients are given for each month and the annual mean. The average of the calculations based on the partitioning into 7, 5 and 3 bins respectively are given. For each of the 51 annual forecast ensembles the probability of the respective exceedance has been determined. The resulting probabilities have been sorted in increasing order and then splitted into equally sized bins and the last bin has

been filled up with the extra years. The observational data have been sorted accordingly and the probability of the quantile exceedance of the observations has then been determined for each bin. The probabilities p, q of model and observational data that the exceedance event occurs and (1-p) and (1-q) that it will not occur then form the basis for the relative entropy or KL divergence for that bin (Eq. (15)). The median and upper quartiles are determined separately for model and reanalysis. The number of bins has been chosen to come close to the proposition of Harremoes (2014) that at least 10 members should be

in the bins. To reduce the influence of the bin positions estimates of 3 different bin partitions have been averaged. In the upper right figure the Pearson anomaly correlation and *ANOVA* are shown for each month and the annual mean (red lines) and the corresponding GCC and GAC curves for median (green) and upper quartile (blue) exceedance. The three curves for correlation and GCC (open circles) allow for the first time a direct comparison of the absolute values of resolution of continuous and exceedance data. Here the highest resolution is found for the continuous data with the correlation and the next

highest value is for the upper quartile exceedance. The lowest resolution is found for the median exceedance. These results follow intuition because most information is in the continuous time series and median exceedance contains a lot of flaws due to the flatness of the pdf near the median at least if the continuous pdf is close to Gaussian. Larger correlation/smaller MSE than *ANOVA*/ensemble spread indicates $ESS > 1$ and thus overdispersion for all months. The balance for the median exceedance is quite close, while the upper quartile exceedance is slightly underdispersive. In terms of the *ESS* also calculated for GCC

and GAC the discrepancy from 1 is largest for correlation and *ANOVA* because here the absolute correlations and *ANOVA* are higher the *ESS* penalizes small deviations much more than for lower absolute values as becomes clear from Fig. 1.





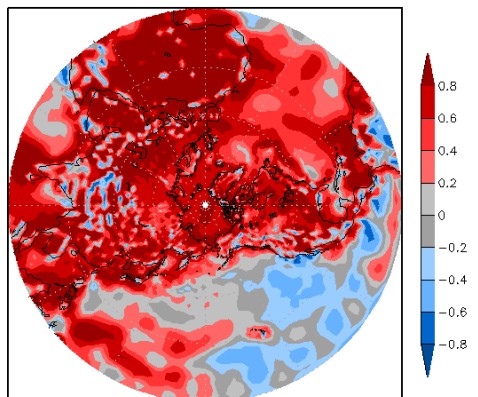
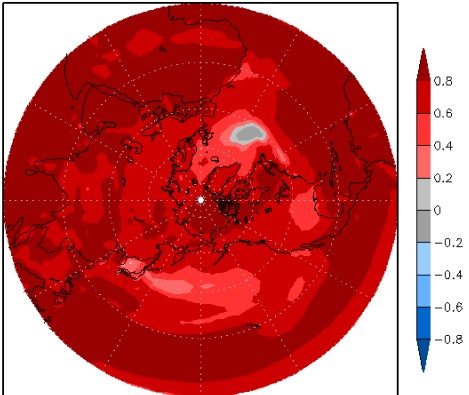

**Figure 6.** EOF1 of the correlation matrix of TAS over the Northern Hemisphere. Left from ERA-40/ERA-Interim data 4 year running averages and right from 45 MiKlip runs lead years 2-5. The model pattern is based on the ensemble mean data.

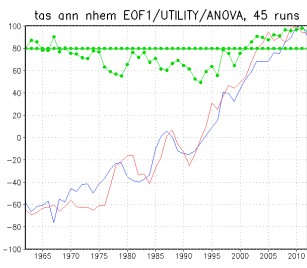
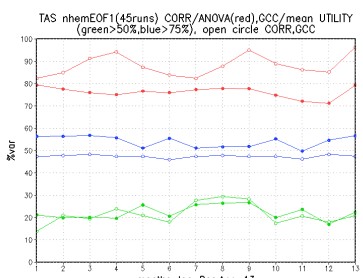
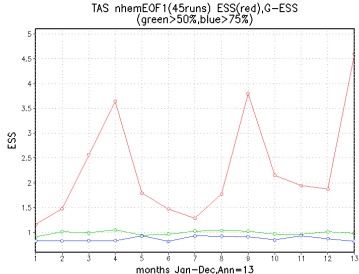

**Figure 7.** Left: TAS EOF1 expansion coefficients (red/ERA-40/ERA-Interim, blue/MiKlip ensemble mean). The green curves are the time dependent *UTILITY* determined from the projection coefficients of the ensemble mean and the 45 runs. The mean *UTILITY* and the *ANOVA* are practically identical. Middle: Correlation and *ANOVA* of TAS EOF1 (red) and corresponding GCC and GAC for 0.5 (green) and 0.75(blue) quantile exceedance. Open circle curves are for correlation and GCC. Right: The *ESS* from *ANOVA* and correlation and from the pseudo correlation and sharpness values GCC and GAC for the quantile exceedances.



## 9 Conclusions

Based on information theory calibration in terms of sharpness and resolution of forecast ensembles can be done with the same
yardstick for continuous data and exceedance probabilities. MI is a measure of resolution in this context. It is a function of
correlation only for Gaussian data. Here it is shown that the average *UTILITY* a measure of sharpness for single forecasts
is close to the ratio of ensemble mean to mean ensemble variance which is the analysis of variance with time as treatment
(*ANOVA*). This leads to the fact that the *ESS* (Palmer et al., 2006; Keller and Hense, 2011) is a measure of calibration if based
on scaled variables. In this form it is a function of correlation and *ANOVA*. We showed that the optimal *ESS* then is equal to
the optimal RPC recently proposed by Eade et al. (2014). Deviating from the RPC the *ESS* incorporates the absolute values
of sharpness and correlation and thus the fact that for large spread the model information content is lost and the statistical
relation of spread and skill is again optimal that means the *ESS* is close to one. If the ensemble spread of a system is very
large and the *ANOVA* thus very low, the mean (scaled) ensemble spread and the MSE between model and observation are close
to one. Thus the *ESS* of scaled variables contains the same information as the rank histogram and the reliability diagram. For
exceedance data the resolution term of the divergence score (Weijs et al. (2010), Ahrens and Walser (2008)) is equal to the MI.
The sharpness is again calculated from the mean *UTILITY*. Here it is calculated in a manner adjusted to the resolution term
of the divergence score measuring the mean distance between single forecast ensembles collected in bins and model climate.
Application of the methods to TAS medium range climate forecasts from the MiKliP project revealed that the calibration of
single grid point data is rarely optimal and that the *ESS* of the MiKlip hindcast ensemble of 45 runs tends to be overdispersive
in regions of relatively high correlations while it is underdispersive in other regions. Overdispersion can only occur with high
correlation (Fig.1) and is indeed observed in TAS retrospective hindcasts of lead year 2 (Fig. 2). This overdispersion is however
largely reduced for the historical runs (Fig. 3) although the correlations are similar. We suspect therefore that the model drift
adds noise to the reforecasts and thus to overdispersion and $ESS > 1$. Underdispersion which is the dominant phenomenon
implies either model flaws or too small spread initialization. The combination of over- and underdispersion in the same fields
could be an indication of model or initalization data flaws. The analysis of *ANOVA* as a function of lead time shows how
anomalies in the initial conditions of the ocean, namely with ORAS4 and GECCO2, can expand over large areas especially
over the southern ocean and finally dominate the model response in large regions. Calibration i.e. sharpness and resolution
analysis for continuous and exceedance data are put into one diagram for the analysis of northern hemisphere TAS EOF 1
which dominates the model and observed low frequency evolution. Even the *ESS* is computed for the exceedance data. The
results support the intuition in that the correlation/GCC is largest for continuous data for which the information content is
largest, followed by the exceedance of the larger (upper quartile) quantile. The median exceedance shows lowest GCC since
the results are dominated by many flaws due to the flatness of the continuous pdf near the median.

*Code and data availability.* Code and data availability. The ERA40 and ERAinterim reanalysis used in this study are freely accessible
through the ECMWF portal after registration. The MiKlip data used for this paper are from the BMBF-funded project MiKlip. Model data of



the described predictions is made available at Climate and Environmental Retrieval and Archive (CERA), a long-term data archive at DKRZ.
The data analysis tools are available at ftp://ftp.meteo.uni-bonn.de/pub/rglowie/vecap-ESS.tgz

*Competing interests.* The authors declare that they have no conflict of interest.

*Acknowledgements.* We acknowledge the funding from the Federal Ministry of Education and Research in Germany (BMBF) through the
research program "MiKlip" (FKZ 01 LP 1520A,B). We would like to thank ECMWF for providing the ERA-Interim and ERA-40 data.



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
