# Peer review of "Common metrics of calibration for continuous Gaussian data and exceedance probabilities"

_Geoscientific Model Development, 2018_

## Referee Comment (RC1) · Anonymous Referee #1 · 16 Nov 2018

**General Comments**

Probabilistic forecasts have recently received much attention across the disciplines. Calibration and sharpness are two common criteria in order to judge the performance of a probabilistic forecast. The present paper develops metrics for calibration and sharpness that (i) relate to information theory, and (ii) apply to continuous and discrete variables alike.

While I am sympathetic with the paper's topic and research question, the paper is not very accessible at present, and there is an important issue regarding the standardization of forecasts (and observations) proposed in the paper. Some specific comments follow.

**Specific Comments**

- While I agree that the paper's objective is interesting, this objective should be better motivated in the introduction, e.g. by mentioning practical situations in which a joint evaluation of continuous and discrete variables is of interest.

- The ensemble-spread score (ESS) at Equation (1) is central to the analysis in Section 2. It seems important to highlight that the ESS is *not* part of the family of (strictly) proper scoring rules (see e.g. Bröcker, 2009, and the references therein). Proper scoring rules are decision-theoretically motivated tools which set the incentive for a forecaster to state what they think is the true distribution of the predictand. They have become a standard tool in recent years, and are probably the concept that many readers have in mind when reading about a 'score'. There are several differences between the ESS considered in the paper and proper scoring rules. First, the ESS is a tool for assessing the performance of a given forecast method, whereas scoring rules are used to compare two or more forecast methods. Second, adopting the paper's notation, the ESS is defined for an entire panel of forecasts $\{Y_{ij}\}$ (with $i = 1, \ldots, nrun$ denoting model runs and $j = 1, \ldots, J$ denoting time) and observations $\{X_j\}$. By contrast, proper scoring rules are defined for an ensemble/observation pair at a given date, such as $\{Y_{i1}\}/X_1$ corresponding to date $j = 1$. Third, the ESS attains its optimal value at one; smaller and larger values indicate a worse forecast, but it is not formally clear whether, say, a value of $0.8$ is better or worse than a value of $1.2$. By contrast, proper scoring rules attain their best value at zero, with larger values corresponding to worse forecasts. Finally, the ESS used in the paper is not to be confused

Interactive
comment

with the error-spread score of Christensen et al. (2015) which is a proper scoring rule based on the first three moments of a forecast distribution.

- The paper proposes to consider the ESS for *standardized* forecasts and realizations, as detailed at Equation (5). While I appreciate the simplicity of the characterizations that follow from standardization (see Equation 7), the comments on P5 highlight an important drawback of the methodology: a noninformative forecast ensemble that is drawn from some arbitrary distribution (such as a standard normal) which is the same at each date $j$ will attain $ANOVA = 0$, as well as the best possible ESS value of one. Due to the standardization step, it is not even necessary for the ensemble to be correctly dispersed. Without the standardization step, the ESS could not be tricked so easily, in that it would at least be necessary for the mean model spread $\sigma_e^2$ to equal $MSE$, c.f. Equation (1). It would hence seem important to provide a more detailed motivation of the standardization step. (On P2, L27 the paper mentions that standardization is necessary in order to interpret ESS as a measure of calibration. This argument is not clear to me, and should be elaborated.)

- The presentation in Section 4 is unclear and should be improved.

  - In Equations (15) and (17), relevant notation ($D_{KL}$, $\hat{X}$, $Y_k$, etc.) is not defined.
  - The stated interpretation of Equation (19) does not become clear at present. To explain the equation, it is necessary to note that $GCC^2/GAC^2$ is the discrete analogue of $CORR^2/ANOVA$. The case $GCC^2/GAC^2 = 1$ then corresponds to $CORR = \sqrt{ANOVA}$, which corresponds to a flat rank histogram by Equation (8). Similar analogies apply when $GCC^2/GAC^2$ is either smaller or greater than one.
  - Relating to the previous comment, the format and wording for Equations (8) and (19) is inconsistent and should be streamlined.

**Minor Comments**

- There are some formal inconsistencies in the paper's citations, see e.g. "(von Storch and Zwiers, 2001)" versus "DelSole (2004)" on P6.

**References**

Bröcker, J. (2009): "Reliability, sufficiency, and the decomposition of proper scores," *Quarterly Journal of the Royal Meteorological Society*, 135, 1512-1519.

Christensen, H., I. Moroz, and T. Palmer (2015): "Evaluation of ensemble forecast uncertainty using a new proper score: Application to medium-range and seasonal forecasts,", *Quarterly Journal of the Royal Meteorological Society*, 141, 538-549.

Gneiting, T. and M. Katzfuss (2014): "Probabilistic forecasting," *Annual Review of Statistics and Its Application*, 1, 125-151.

---

## Referee Comment (RC2) · Anonymous Referee #2 · 23 Nov 2018

**Report on the manuscript:**
*Common metrics of calibration for continuous Gaussian data and exceedance probabilities*
by Rita Glowienka-Hense, Andreas Hense, Thomas Spangehl, and Marc Schröder

The paper is concerned with the evaluation of ensemble forecasts for Gaussian data or categorical data. It appears that the authors are seeking to introduce univariate summary measures of calibration and sharpness of ensemble predictions that yield comparable values across Gaussian or categorical outcomes. I have several severe concerns with the paper which I am listing below.

**Detailed comments:**

1. The motivation, goal and results of the paper are unclear. Why is it important to compare calibration of predictions for Gaussian outcomes or for categorical outcomes on the same scale? Why can this not be achieved by using a proper scoring rule that applies to both, continuous and discrete outcomes such as the CRPS?

2. The paper suffers from a number of mathematical inconsistencies such as missing assumptions and definitions, or, overly simplistic and, therefore generally incorrect statements. Some examples are

   - Equation (8) suggests that rank histograms can either be flat, inverse U-shaped or U-shaped. However, there are many other shapes possible which are not just due to finite samples. It should be clearly stated and proved under which implications are intended in equation (8) and under which conditions they hold.

   - Equation (9) is introduced as a definition but then it is an inequality. This is particularly confusing in view of the statement on p.16,l.7-8. How is the RPC defined? What do you mean by "the optimal ESS is then equal to the optimal RPC"? Are they just both equal to one for calibrated predictions?

   - p.6,l.4-5: Starting with ensemble forecasts, there are many ways to derive predictive densities $p$. Similarly, the climate pdf can be estimated in numerous ways from the observations. What do you mean here?

3. The paradigm of Gneiting et al. (2007) to "increase sharpness subject to calibration" is not appropriately applied by the authors. Gneiting

et al. (2007) rigorously define both concepts and sharpness refers to the forecasts only, whereas calibration ensures statistical compatibility between forecasts and observations. This is contradictory to the statement on p.2,l.7–8: Calibration in the sense of Gneiting et al. (2007) is not a balance between sharpness and resolution. A calibrated prediction can be very sharp or not sharp at all.

4. In relation to my previous comments, I have severe reservation to speak of an "optimal" value of ESS being 1. It is not based on a proper scoring rule, and the authors do not give rigorous arguments of what is meant by "optimal" here an why (and under which conditions) this "optimum" is achieved *if and only if* $ESS = 1$.

5. In line with my last comment is the following issue: On p.16,l.12 the authors state that "the ESS of scaled variables contains the same information as the rank histogram". Firstly, if this is true then this is a strong argument against using the ESS to assess the quality of ensembles in terms of calibration *and* sharpness because the rank histogram does not assess sharpness. This can be shown rigorously and examples of this nature are provided in Gneiting et al. (2007). Secondly, in this broad generality, I believe that the statement is false, see my previous comment on equation (8) above.

---

## Author Comment (AC1) · 19 Dec 2018

Thanks to the reviewer for reading our paper and providing some helpfull comments. Especially we will try to motivate the use of similar scoring rules for continuous and categorical data as well as the standardization of the ESS score and underline the different model problems for ESS below and above 1.

**1 Replies to specific questions**

**1.1 Why is the evaluation of joint and discrete variables with the same score of interest?**

Comparing continuous correlation of e.g. temperature between forecast and observation with transformed mutual information from exceedance probabilities of quantiles can reveal where the overall correlation comes from.

**1.2 ESS is not part of the family of strictly proper scoring rules. It attains its optimal value at one. Differing from the ES**

Assuming that the discussed time series have Gaussian distributions, i.e. the skewness terms are zero, the error spread score (ES), which is proper, contains the same 2 terms as the ESS.
Thus if the ESS is 1 the ES is zero.
The ESS is not a complete scoring rule in itself.
It is a measure of reliability. It measures whether the observations can be seen as ensemble members of the forecast.
The associated measure of resolution is the correlation and the anova with time as treatment is the associated measure of sharpness.

**1.3 ESS attains optimal value one. Is 0.8 worse or better than 1.2?**

If the ESS is equal to 1, then the ensemble spread indicates the model uncertainty.
Values of the ESS below and above one should be interpreted differently. Too low ensemble spread and thus too sharp forcast ensembles may well be a problem of model physics and only for very short term forecasts a problem of too small spread of the initial ensembles. ESS values above 1 indicate additional noise in the model. Thus theoretically values below 1 are more problematic than an ESS above one. We will underline this point in the paper.

**1.4 ESS does not consist of ensemble observation pairs at given dates**

The denominator of the ESS consists of the average over all forecasts of the pairs of ensemble mean at time j and the corresponding observations at time j with the respective distances between ensemble mean and ensemble members at a specific time step.
The ESS is created from the same two variables namely mean square error between ensemble mean and observation and ensemble spread as the error spread score ES which is a proper score. We will describe this point more clearly in the paper.

**1.5 Why standardization**

The ESS here has been performed with standardized variables. Thus the effects of too large/low model variance or bias - which could be remedied by post processing i.e marginal calibration - have been eliminated.
This helps to separate the marginal calibration from the reliability issue. This is in line with the argumentation of Bröcker (2009), who is dealing with calibration methods. He proposes to take ensembles as a source of information only. Moreover the standardization is a basic feature of regression analysis:

$$\hat{Y} = \bar{Y} + \beta X, \text{ where } \beta = \frac{\sigma_y}{\sigma_x} CORR(X - \hat{X})$$

$$\frac{(\hat{Y} - \bar{Y})}{\sigma_y} = CORR \frac{(X - \bar{X})}{\sigma_x}$$

which is one way of taking the information from a forecast. Here X refers to the ensemble mean of the forecast and Y to the observation. CORR is thus the regression coefficient in case of marginal calibration. The inference of our calculations is that in the ideal case of an ESS=1 the ensemble spread is equal to one minus the squared correlation.

**1.6 Uninformed forecast with ANOVA=0 has optimal ESS=1**

We are aware of the fact and it is discussed in the text that an EPS can be perfectly reliable without being sharp. If a forecast is not sharp then every observation fits into the forecast ensemble. This is why there is a need for at least two of the variables - reliability, resolution and sharpness - to describe the performance of the forecast.

If the skewness is zero, the ES=0 in case the ESS=1. This is also true for ANOVA=0. It can be directly seen because the terms are the same.

**2 Page 2 line 27 because only then the score is a measure of calibration as shown here..**

Here probabilistic and exceedance calibration is meant and for claritiy will be replaced by the term reliability.

**2.1 notation section 4**

We will explain the notation in section 4. Further we will eliminate citation inconsistencies.

---

## Author Comment (AC2) · 19 Dec 2018

We thank the reviewer for discussing our manuscript. We hope that we can convince him of the benefit of our methods.

**1 Replies to specific issues**

**1.1 The aim univariate summary measures..**

- The *first* main result was to show the relation of the ESS to basic well established scores correlation and ANOVA with time as treatment.
  The standardization we introduced is also called marginal calibration and is a standard procedure in regression analysis. For standardized variables the correlation coefficient is the regression coefficient.
  Assuming Gaussian distribution of the variables the same terms are involved in the ES score (Christensen et al 2015 ) - to which we were made aware by the other reviewer - if used with standardized variables and assuming Gaussian distributions and thus zero skewness.
  This standardization leads to a reliability measure that is rid of marginal calibration errors.
  The result shows further that the optimal ensemble spread is equal to $1 - CORR^2$ and equivalently the RPC=1. The RPC is defined and discussed in Eade et al, 2014. We simply repeat this discussion.

- The *second* result was that we could show that the ANOVA ratio is very close to the mean *utility* defined by Kleeman (2002). This connects ANOVA analysis and relative entropy.
  The mutual information (MI), which is a special integrated relative entropy between the joint and the marginals of two variables, is directly related to correlation this comes from the literature. Together this shows that the classic tools used for the analysis of forecast ensembles are directly related to relative entropy.

- The *third* concern was to use a similar method for categorical forecasts.

**1.2 Why not CRPS?**

The CRPS has been shown to be beneficial for evaluating ensemble prediction of financial portfolios where always the complete pdf matters. In case of Gaussian distributed time series it can happen that an EPS with incorrect marginal calibration like too large variance but medium correlation has a larger CRPS than a second ensemble prediction system (EPS) with nearly zero correlation but well

adapted variance. Thus the uninformed second system wins in comparison to the informed first system.

If we perform marginal calibration then the only thing that remains from the orignal ensembles is the relative spread (1-ANOVA) and the ensemble mean.

The CRPS is important to evaluate large ensembles of financial portfolios because here always the complete pdf materializes. The CRPS is also well suited to compare two mean climate projections including their uncertainty.

**1.3 The rank histogram can have different forms others than U-shaped and inverse U-shaped**

The asymmetric forms of the rank histogram would be due to positive or negative mean biases of the EPS but in our case the data have been standardized (= marginal calibration see above) before the analysis. The rank histograms of these data will be generally U-shaped or inverse U-shaped. Overlays of these shapes are imagineable in case the ensemble sharpness varies with the initial state of the prediction. We guess that the time series of such systems will not have Gaussian pdf as is assumed here.

**1.4 Under which conditions eq. 8 holds ...**

We will further underline that also the relations to the rank histogram analysis in eq. 8 hold for standardized Gaussian variables. If the ESS for these scaled variables is equal to one or equally the ES be zero the distance of the ensemble members from the mean is the same as the mean distance between the observation and the ensemble mean. Thus the observation behaves like an additional ensemble member and therefore the rank histogram would be flat.

On the other hand if the ESS is less than one the ensemble members are generally closer to the ensemble mean than the observations. As we have scaled the variables to have zero overall mean, this means that the rank histogram is U-shaped. Analogously in case of an ESS greater than one the smaller distances of the observations from the ensemble means leads to an inverse U-shaped rank histogram.

**1.5 How is the RPC defined?**

The ratio is defined by Eade et al 2014 as a lower bound for the actual ratio of predictable components (RPC) which might be improved by future model developements. We took the definition directly from the paper. However the authors did not use the term ANOVA for the ratio of mean ensemble spread to total spread. They claim that the ratio $\frac{CORR}{\sqrt{ANOVA}}$ should ideally be equal to one without giving any proof.

**1.6 Are the ESS and the RPC just equal to one for calibrated predictions?**

The ESS can also be equal to one if the variables are not normalized. If this happens for model and the observational data with equal marginal calibration then the model ensemble is indeed reliable. On the other hand differences in the marginal calibration of observations and model can lead to an $ESS = 1$ but without having a reliable forecast ensemble.

Reliability of an EPS implies that the ensemble spread is equal to the mean square errror between observations and ensemble means and thus $ESS = 1$. In case of standardized variables this further implies that the correlation is equal to the square root of the ANOVA. This means that the claim of Eady et al. (2014) is equivalent to reliability of an EPS after standardization of the data.

**1.7 Does the RPC need calibration**

The ratio of correlation to the square root of ensemble mean to total variance depends on standardized variables. Thus the marginal calibration is inherent.

**1.8 How have the predictive densities been derived?**

Assuming that both forecasts and observations are Gaussian distributed an overall mean has been determined and the variance is an average variance with respect to that mean. The ensembles here do not show systematic differences. The pdf at a special forecast time is directly determined from the ensemble mean and the ensemble variance.

**1.9 The paradigm of Gneiting et al. (2007) "increase sharpness subject to calibration" is not appropriately applied by the authors.**

The sharpness measured here with ANOVA is an attribute of the forecasts only as is demanded by Gneiting et al (2007). It is calculated without any reference to observations, it can be generated right after the EPS prediction is available.

Measuring the reliability with standarized/marginally calibrated variables is intended to give an indication whether the sharpness - measured with ANOVA - is indeed associated with calibration (exceedance + probabilistic)/reliability.

**1.10   A calibrated prediction can be very sharp or not sharp at all**

The ESS analysis includes that forecasts can be reliable/calibrated (beyond marginal calibration) but not sharp at all. If the model sharpness/ANOVA is zero then the forecast is reliable or probabilistic and exceedance calibrated in any case (eq 8). For low correlation the model sharpness/anova should be correspondingly low then the forecast is also calibrated(exceedance + probabilistic)/reliable. Therefore as sharpness is increased the reliability/calibration(beyond marginal calibration) can only be hold ($ESS = 1$) if the resolution/correlation increases accordingly. Thus the reliability/calibration (beyond marginal calibration) is indeed a balance between sharpness and resolution.

**1.11   Why is the optimal value of ESS=1**

Optimal is meant in the sense that the ensemble spread is equal to the mean square error between observations and ensemble means. The same is demanded in the ES score of Christensen et al (2015) and citations therein in case of Gaussian distributions and thus zero skewness. Your co reviewer pointed us out to this article, we will cite it in our revised version. They have the same two terms - without standardizing the data - but take the squared difference. Thus the ES of standardized variables should be zero in case the ESS is one. The ES is a proper scoring rule. If you perform the same transformations the ES equally only depends on correlation and ANOVA. From the squared difference it can however no longer be determined whether the EPS is over- or underdispersive which we think is important.

**1.12   Rank histogram does not assess sharpness**

The rank histogram is as the ESS a measure of reliability of the forcast ensemble. In case one uses marginally calibrated data also for the rank histogram a U/inverse-U-shaped rank histogram is indicative of under/over-dispersion. This means on the one hand that the data must have also resolution because otherwise the rank histogram of a marginally calibrated data set of observations and prediction ensemble would be flat. On the other hand the underdispersion/overdispersion is indicative of too large/low sharpness of the forecasts compared to resolution. The rank histogram gives however no quantitative information for this missing balance and is no absolute measure of sharpness. The latter depends on the forecasts only. Such numbers are given by the triplet of ESS, correlation and anova. The relation between rank histogram and ESS only holds for Gaussian distributions and if standardization/marginal calibration is performed.